# Indoor Positioning System Based on Fuzzy Logic and WLAN Infrastructure [note 1]

**DOI:** 10.3390/s20164490

**Published:** 2020-08-11

**Authors:** Jaromir Hrad, Lukas Vojtech, Martin Cihlar, Pavel Stasa, Marek Neruda, Filip Benes, Jiri Svub

**Affiliations:** 1Department of Telecommunications Engineering, Czech Technical University in Prague, Technická 2, 166 27 Praha 6, Czech Republic; hrad@fel.cvut.cz (J.H.); vojtecl@fel.cvut.cz (L.V.); m.cihlar99@gmail.com (M.C.); marek.neruda@fel.cvut.cz (M.N.); 2Department of Economics and Control Systems, VSB-Technical University of Ostrava, 17. listopadu 2172/15, 708 00 Ostrava-Poruba, Czech Republic; filip.benes@vsb.cz (F.B.); jiri.svub@vsb.cz (J.S.)

**Keywords:** fuzzy logic, indoor positioning, localization, wireless networks

## Abstract

This paper deals with the issue of mobile device localization in the environment of buildings, which is suitable for use in healthcare or crisis management. The developed localization system is based on wireless Local Area Network (LAN) infrastructure (commonly referred to as Wi-Fi), evaluating signal strength from different access points, using the fingerprinting method for localization. The most serious problems consist in multipath signal propagation and the different sensitivities (calibration) of Wi-Fi adapters installed in different mobile devices. To solve these issues, an algorithm based on fuzzy logic is proposed to optimize the localization performance. The localization system consists of five elements, which are mobile applications for Android OS, a fuzzy derivation model, and a web surveillance environment for displaying the localization results. All of these elements use a database and shared storage on a virtualized server running Ubuntu. The developed system is implemented in Java for Android-based mobile devices and successfully tested. The average accuracy is satisfactory for determining the position of a client device on the level of rooms.

## 1. Introduction

There are many situations that require a more or less precise localization of mobile units—from applications in military to healthcare, from storage management to road navigation—each of them having specific conditions and requirements.

The hospital environment is the most common example used to illustrate the issue of indoor localization. A hospital complex is generally a large and structured area where a patient or hospital equipment needs to be localized. In practice, this could mean that, for a specific building, the positions of all present persons are recorded. In case of an accident (e.g., fire), the rescue team can use the latest localization data to predict how many people are inside the building and also where these people were located when the fire broke out.

The demands for the deployment of an indoor localization system is constantly increasing. Thanks to the Internet of Things (IoT) and its ubiquitous connectivity, it can provide useful information and a wide range of services. Different techniques, wireless technologies and mechanisms have been proposed by many researchers to provide indoor positioning systems in order to improve the services provided to their users.

This article is an extended version of a conference paper [1], in which an experiment proving the capabilities of standard Wi-Fi infrastructure for independent positioning was described.

## 2. Localization in a WLAN Environment

### 2.1. Localization in the Environment of Buildings

Probably the best-known term in the field of localization is the Global Positioning System (GPS), which is used practically everywhere—for example, in car navigation, for hiking, in shipping, and aviation. However, all these applications have one thing in common: they are all used for outdoor localization. The ideal case, therefore, is when there is direct visibility of at least three to four satellites from the ground GPS station. This, however, cannot be achieved in most indoor areas of buildings. Due to the fact that signal propagation is the key parameter of GPS, the resulting accuracy of the localization is very low inside a building, if there is any connection between the ground station and the satellite at all. Therefore, other available technologies must be used for indoor localization.

Generally, localization systems can be divided into two categories: radio frequency (RF) and non-RF (ultrasound, infra-red light, laser, and others). The main advantage of RF as opposed to non-RF is its possible application in an environment without direct visibility, which is practically required by all non-RF localization systems.

The RF localization systems for indoor use (Indoor Positioning Systems) utilize various wireless technologies, such as Bluetooth, Radio Frequency Identification (RFID), Wi-Fi, and others. Most of these technologies, however, were not designed specifically for localization purposes and, therefore, it is necessary to modify or extend these systems for such use. For example, Wireless Local Area Networks (WLANs) are relatively expensive to build and the respective devices are too large to mount onto small objects. These technologies provide only “estimated” locations of objects, calculated, e.g., by triangulation based on data from sensors.

The usage and deployment of Real-time Locating Systems (RTLS) in a dynamic environment is affected by various problems. For example, the GPS signal is too weak to work inside a building, and the accuracy of localization decreases with increasing density of the environment where signals interfere with each other. Ultrasound may provide very high accuracy, but it requires configuration with direct visibility as it can penetrate objects only with sufficient power. Similarly, visual RTLS is affected by reflections and optical environment variations. They can also “lose” the monitored object if it is moving very quickly or if it is “hidden” in shadow. RF technology, Ultra-Wideband (UWB), and RFID require an LAN connection for accurate localization. Disconnection from the LAN causes a significant decrease in accuracy and may become a problem in larger areas where cabling is required, which makes the whole system more expensive.

Table 1 summarizes a basic comparison of the main technologies that can be used for an Indoor Positioning System (IPS).

When evaluating positioning systems, it is appropriate to consider their accuracy, complexity of deployment, costs, and scalability. One of the possible principles is the evaluation of the Wi-Fi signal level. In general cases, positioning accuracy is between 3–5 m in a typical indoor environment. Due to the fact that most mobile phones have the Wi-Fi function, this technology is easily portable and its cost is low. However, the installation of a Wi-Fi localization system is complicated and time-consuming. Before such a system can be deployed, it is necessary to create radio maps using the fingerprint method. The complexity of engineering work significantly increases in larger indoor areas. This is the reason why, in the retail sector, Apple iBeacon (Bluetooth 4.0) is used instead of Wi-Fi. UWB technology can reach an accuracy of up to 20 cm, and therefore UWB is appropriate for places where high accuracy is required.

The accuracy of various kinds of localization systems is different, which is given by the localization detail level. The list of individual systems is as follows [3]:zone-level localization—the zone where a user is located and is identified; the zone can be defined as a room, more rooms, or part of a building, for example:room-level localization—the room where the user is located and identified;partial room-level localization—for large rooms, it is possible to determine, for example, the part where a user is located;localization on the basis of passage—the position of the user is determined as he/she passes through a control gate, which can be equipped by, e.g., doors with sensors;localization in reference to a point—the output of the localization is the position of a user in reference to a known point;precise localization—the position of the user is given by precise coordinates in a given area.

The accuracy of a system is also based on the abovementioned levels of localization, which express how much the system deviates in its determination of a user’s position from the actual one. This means that, in zone-level localization, which this work focuses on, results with centimeter-level accuracy cannot be expected.

A summary of the accuracy of the basic IPS technologies is given in Table 2. The results of the studies suggest similar accuracy to commercial RTLS. New techniques and algorithms increase accuracy, and in some cases, such as for RFID, Montaser and Moselhi [4] report higher accuracy than commercial solutions—an average deviation of only 1 m in a person’s location compared to 2–3 m.

#### 2.1.1. Localization Methods

Localization methods are based on measured signal characteristics. These can be time latency (from sending the signal to its reception) or the angle at which the signal is received by the receiving antenna or the level of the received signal.
**Time of Arrival (TOA)** is a method that evaluates signal propagation times between a transmitter and a receiver. The time is calculated from a fixed time “0” (identical for both devices). Therefore, it is necessary to precisely synchronize the devices, i.e., the network element (Access Point, AP) and the localized device (smartphone, Wi-Fi adapter, etc.) A major disadvantage is the sensitivity to different times of signal propagation. Especially in the indoor areas of buildings, signal propagation is affected by obstacles. This causes the multipath propagation phenomenon, i.e., the signal can be received from different directions with different levels, but also with different delays. This method is therefore suitable for the outdoor environment where the signal does not interfere with so many objects. The position calculation itself is done with the help of the trilateration algorithm.**Time Difference of Arrival (TDOA)** is based on a similar principle; however, it is not necessary to synchronize the localization devices. The device to be localized sends information, which is received by all access points. The position of the device is calculated from the different times of message arrival to the respective APs and their known position. This method is also very sensitive to signal propagation time. The trilateration algorithm is used here as well.**Angle of Arrival (AOA)** is a method based on measuring the angle at which a signal is received with respect to a reference point. This method is barely usable for indoor localization, because the signal propagates in multiple directions. Therefore, the stability of direction from which the signal arrives cannot be guaranteed. The triangulation algorithm is used to calculate the position.**Received Signal Strength Indicator (RSSI)** is a method based on measuring the signal level at a given point in a building. Due to the fact that the signal level and distance change both naturally and due to the effects of the environment, the outputs of such measurements are the RSSI characteristics of the entire environment. This characteristic is different for each AP. The characteristic from the training phase can be compared with a specific measurement in the localization phase. There are more algorithms for the calculation of positions for this method. It can be the method of the nearest neighbor, an improved variant of the nearest neighbor (kNN). Neural networks, genetic algorithms, data clustering methods, and other algorithms, which can compare two different types of measurement methods and find possible similarities, may be used.

#### 2.1.2. Fingerprinting Method

The positional fingerprint method [6] is probably the best solution for localization inside buildings. The method consists of two phases. The first one is called “offline” (sometimes also the training phase). The second one is called “online”, which can be described as the phase of self-localization. The main purpose of the first phase is to create a training data set, which characterizes signal coverage of the given area. The creation of data is performed by determining several measurement points where the RSSI values are scanned from surrounding access points. The scanning is done by an administrator or a designated person (not by a localized user who need not be bothered). The entire area is traversed in this manner, and the resulting training set is then compared with RSSI values, which are scanned during self-localization.

There can be any number of RSSI values (i.e., the number of APs from which these values are taken), however, there should be at least three of them; the ideal number is four to six. The higher the number of RSSI values, the higher the variability of results; it is more difficult to distinguish the correct result in the data. The RSSI method described above is used to calculate the position. The localization system can therefore be generally characterized by a number of elements, by the device which creates the training data set, by the device (identifier) which is being located, the method of localization, and by the server which constitutes the core of the system.

### 2.2. Related Work

Over recent decades, many researchers have dealt with indoor localization techniques and proposed their indoor positioning systems, architectures, algorithms, and so on. Some of them are time-consuming, labor intensive, and easily affected by environmental issues.

Chen et al. [7] described in their study a localization solution that uses Prior Information (LuPI). The key idea of their solution is that human motion can be distinguished and recorded based on the radio information (e.g., RSSI deviation between different positions) and a pedometer. LuPI utilizes the RSSI and a sensor-based pedometer to build an RSSI variation space as the prior information. Then, based on this prior information, LuPI estimates the location of a mobile node. Based on their results, the average localization error is 5.9 m in a corridor, 1.4 m in a big room, and 1.9 m in a small room. We can conclude that the localization accuracy of LuPI is on the room level.

Wu et al. [8] presented a method called Wireless Indoor Logical Localization (WILL). The localization solution is based on two phases: training and serving. In the first phase, traditional methods involve a site survey process, in which engineers record the RSSI fingerprints at every position of the area of interest, and build a fingerprint database. Next, in the serving phase, when a user sends a localization query with its current RSSI fingerprint, the algorithms retrieve the fingerprint database and return the matched fingerprints, as well as their corresponding locations. The main idea of their method combines Wi-Fi fingerprints with user movements. Fingerprints are divided into different virtual rooms and a logical floor plan is constructed accordingly. Localization is achieved by finding a match between the logical and physical floor plan. Their solution achieved an average room-level accuracy of 86 percent.

Yang et al. [9] have proposed a calibration-free indoor localization scheme that uses existing Wi-Fi infrastructure. The proposed radio map building and localization techniques are based on the general relationship among the RSSI values for the individual APs. Based on the provided results, the error does not exceed 2 m for a hallway and 4 m for a laboratory.

Farshad et al. [10] have examined the impact of various aspects underlying a Wi-Fi fingerprinting system. They investigated different definitions for fingerprinting and location estimation algorithms across different indoor environments, ranging from a multi-story office building to shopping centers of different sizes. The results show that the fingerprint definition is as important as the choice of location estimation algorithm and there is no single combination of these two that works across all environments or even all floors of a given environment. Farshad et al. compared three different deterministic techniques (including the often-used Euclidean distance-based nearest neighbor method) with two probabilistic techniques that use Gaussian and log-normal distributions for RSSI modeling. Their results show that the combination of fingerprint definition and location estimation algorithm that provides the best localization accuracy is highly dependent on the environment and even a specific floor within a given environment. They also found that the choice of frequency band (2.4 GHz vs. 5 GHz) and the inclusion of Virtual Access Points (VAPs) has a significant impact on the accuracy of Wi-Fi fingerprinting systems.

Torres-Sospedra et al. [11] discuss the details of the 2017 Indoor Positioning and Indoor Navigation (IPIN) indoor localization competition, the different data sets created, the teams participating in the event, and the results they obtained. In their paper, they compare these results with other competition-based approaches (Microsoft and Perf-loc) and online evaluation websites. The winner of that edition was the UMinho team (score: 3.48 m) followed closely by the AraraDS team (score: 3.53 m). The difference between their scores was only 5 cm, which demonstrates the quality of both competing systems regarding the metric. As mentioned in [11], AraraIPS’s approach to indoor positioning has four distinctive characteristics: it is based on a cartographic paradigm (fingerprinting), it uses a discretization of the predicted floor/building, it is measurement agnostic (i.e., its abstract formulation is not specific to any kind of signal or measurement, such as Wi-Fi, magnetic field, Bluetooth Low Energy (BLE), etc.), and it exploits measurement history. In the following paragraphs, these features will be explained to provide a better understanding of how AraraIPS works. The core of the positioning estimation solution approach used by the winners (UMinho team) is a plain Wi-Fi fingerprinting estimation algorithm based on the k-nearest neighbors classifier. However, the optimum use of this base algorithm in the context of the IPIN 2017 competition required a set of additional data manipulation and estimation processes.

Ferreira et al. [12] presented four ToA-based positioning algorithms that were evaluated under different conditions, e.g., environments with different propagation conditions, static and dynamic targets, and with or without non-line-of-sight (NLOS) influence due to the presence of a human body. Among all the evaluated algorithms, the extended Kalman filter has shown the best results for all performance metrics in the static scenarios and when two anchor nodes are in an NLOS condition for the dynamic tests.

Honkavirta et al. [13] offered a solid theoretical background of WLAN location fingerprinting methods, while our paper is more practice oriented. Due to the difference in RSSIs between devices, the comparison is not sufficient. The Signal Strength Difference (SSD) method was used to eliminate this problem. Instead of processing specific RSSI values from a given AP, the difference between two RSSI values from two access points is processed. They also used a Bayesian filter while we use only Kalman.

Han et al. [14] offers overall insight into the process of the commercial implementation of a WLAN localization system, while they do not offer much information about the principles and algorithms that were used. Practical information about response delay, battery drainage, etc., were discussed in this article.

Different technology (ultra-wideband (UWB) radio and inertial measurements) was used by Ferreira et al. [15], while similar processing techniques (Kalman filtering) were addressed together with trilateration and the Taylor series method. UWB devices needed to be developed while we used the current implementation environment (WLAN APs). The utilization of Inertial Measurement Units (IMU) and pedestrian dead reckoning technology can be quite useful to obtain information about dynamic location changes.

Niu et al. [16] proposed a heatmap-based Wi-Fi fingerprinting method by utilizing a period of history location estimations as an additional input to improve Wi-Fi fingerprint localization in an open space environment. To generalize the impact of an indoor environment, the authors have conducted Wi-Fi-fingerprinting experiments in typical distinctive indoor areas. In their testbed, they used much more APs than we did (51 pcs).

He et al. [17] described, in their study “Comparisons of Wi-Fi Fingerprint-Based Indoor Positioning Methods”, an overview of recent advances in two major areas of Wi-Fi fingerprint localization, advanced localization techniques and efficient system deployment. They also focused on different approaches in the location of the AP regarding the internal layout of the building, like narrow paths, corridors, large indoor open spaces, and room partitioning. As for typical collaborative localization methods, they mentioned Bluetooth, ZigBee, and ultrasonic. They also see combining vision with Wi-Fi fingerprinting localization as an interesting research direction. The best presented methods reached an accuracy of 1–2 m. It is a mainly a survey article that compares Wi-Fi fingerprinting methods and approaches. To provide more precise localization, they also described the utilization of digital cameras and computer vision.

Fuchs et al. [18] showed that the combination of multiple localization methods forms a promising approach for deployment. The ultra-wideband technology appears to be the most suited for indoor localization, as it allows for a good precision even in the presence of obstacles. It is mainly a survey article. During the testing of the described solutions, they involved the Navshoe system, including a very precise accelerometer and a gyroscope affixed to a person’s foot.

Fischer et al. [19] deal with the issue of internal localization with a focus on emergency response scenarios. High temperatures, thick smoke, noise, gusts of air, obstacles, and falling debris hinder the propagation of the radio, ultrasound, and laser signals typically used for localization. They offer an overview of the most common methods and their potential application to emergency responses.

Ferreira et al. [20] provide a very detailed overview and comparison of survey papers on indoor localization. The article is focused on reliable, scalable, affordable, and an autonomous IPS for emergency responders. Furthermore, the survey was expanded by localization methods and techniques used for indoor localization. Their paper presents an overview and requirements for emergency responders and includes a deep comparison of the surveyed IPSs.

### 2.3. Summary of IPS

The previously performed tests with developed applications were based on the fingerprinting method (see [21]) as a relatively accurate type of localization; it is able to determine the precise coordinates in a given area—an average accuracy of 3 m was achieved. For these purposes, the average accuracy of localization is irrelevant, since it does not work in a way that would be expected for a precise localization. This is mainly due to the technology which is used for the localization, i.e., Wi-Fi.

This technology is not suitable for precise localization. Although its accuracy can be improved to 1.5 m, it would either require laboratory conditions, or an unnecessarily complicated algorithm. These conclusions are valid, of course, for the environment of buildings where direct visibility between a terminal and an AP is not achieved. With respect to this fact, the localization target was reevaluated, and the current localization system is designed for zone-level localization.

From a purely practical view, zone-level localization is required in most cases; it is not necessary to monitor whether a patient is lying in bed or moving in the room. It is only important to know the room (zone) where he/she is currently located. Moreover, a better localization success rate is guaranteed, as a zone is much easier to identify than specific coordinates.

## 3. Problems to Solve

### 3.1. The Variety of Wi-Fi Adapters

This problem can be seen when comparing the devices of various manufacturers, but also when comparing different devices from the same manufacturer. Unfortunately, not even Wi-Fi adapters of the same model line are always identical. In the performed measurements, the RSSI value differed in a number of devices, even by 20 dBm in the same place in the area and with the same orientation of the device. The consequence is that the accuracy of localization using the Wi-Fi-based system is entirely degraded. Specifically, it is the denial of the principle itself, which is based on the fact that the first device measures RSSI values (training phase) in reference to the space and the second device (localization phase) compares its measured values with the training phase. Due to the difference in RSSIs between the devices, the comparison is not correct.

The Signal Strength Difference (SSD) method is used to eliminate this problem [22]. The principle of this method is that, instead of processing specific RSSI values from a given AP, the difference between two RSSI values from two access points is processed. Thus, individual differential RSSIs will be defined as Δ1−2, Δ2−3, etc. If the signal is measured from access points, the given measurements will be defined by *N-1* differential RSSI values. In order to achieve the required accuracy of the designed system, five differential values are used, and therefore six access points are needed.

The results from subsequent RSSI measurements are illustrated in Figure 1 and Figure 2, where three devices, measuring signal level at the same time and in almost the same place, are compared. Due to the fact that it is not exactly the same place, there may be deviations in the measurements caused by this measurement error. The diagram, however, serves as an explanation of another problem. In Figure 1, there can be clearly identified a stair-like shape of the values in both Samsung phones in comparison with LG. The stair shape is caused by the fact that, for a certain time, the signal level has the same value. The RSSI data was measured in the interval of a single second, that is, the minimum possible time in which the RSSI values change. In other words, Android OS has a preset system parameter, which defines the interval in which measurements of RSSIs of the surrounding WLAN networks are done. This parameter can be unfortunately different for every single device, either because it is another brand, or it runs a different version of Android. The method by which the application obtains the RSSI values therefore does not initiate the scanning of WLANs in the given area, but only accesses these results, which automatically change depending on the value of the parameter (measurement interval).

This parameter is set to 4 s in testing telephones (Samsung); that is why the RSSI value does not change in the first four seconds, compared to LG, where the value changes every second. Due to the fact that a certain number of values is scanned and then averaged, there were only two different values in 8 s for Samsung, whereas LG had eight values. This disparity of values clearly affects the resulting average. For this reason, it is necessary to calibrate the devices before the measurement. The calibration process consists in one-minute calibration measurements, from which the system parameter of RSSI measurement is determined. The scanning interval of a given device is then adjusted while maintaining the same number of measured values for all devices. In practice, this means that, in one minute, the value of the system parameter is identified—for example, 5 s. The average is calculated from five values; this means that the scanning duration is 25 min. For other devices, it may differ depending on the system parameter. This result also affects the positioning interval of the device but not its accuracy.

### 3.2. Signal Fluctuation

The most serious weakness of localization based on Wi-Fi signal is its variability in time. When it comes to localization inside buildings, the fluctuation is even more pronounced due to multipath signal propagation. Unfortunately, this is a natural phenomenon which cannot be entirely eliminated, and therefore it is necessary to minimize it. The first version of the localization system worked on the basis of measuring several values of the signal level, which were then averaged. It is probably the simplest algorithm for minimizing fluctuations, but it is still largely affected by the extremes of the measured values. This means that the result can often be affected by signal failure, or a very low signal level. For this reason, it is necessary to filter these extremes out of the measured data.

It is definitely a bad practice to completely remove these extreme values. The final value would then be derived from a smaller number of values than desired. The Kalman filter is used to minimize fluctuation in the current system. It is an adaptive filter which describes the states of the discrete system depending on the current measurement. It is based on gradual iteration, where the characteristics of the filter change in each step based on input values. The main benefit of this principle is that historical measurement values do not need to be stored. The filtration itself is recurrent, consisting of two parts—prediction (time update, (1), (2)) and filtration (measurement update, (3), (4), (5)) [23].
(1)$${\hat{x}}_{\overset{¯}{k}}=A{\hat{x}}_{k-1}+Bu_{k-1},$$
(2)$$P_{\overset{¯}{k}}=AP_{k-1}A^{T}+Q,$$

The first part (prediction) provides a new status value ${\hat{x}}_{\bar{k}}$ (mean value of the measurement) and covariance matrix $P_{\bar{k}}$ from the previous state ${\hat{x}}_{k-1}$, i.e., $P_{k-1}$. For the first iteration, the values ${\hat{x}}_{k-1}$ and $P_{k-1}$ must be initialized. Values ${\hat{x}}_{k-1}$ and $P_{k-1}$ are the previous ones, which are filtered to the resulting values based on the second part of the algorithm (filtering). Before the filtering itself, the filter characteristics are updated (${\hat{x}}_{\bar{k}}$,$P_{\bar{k}}$) with a new measured value *k*.
(3)$$K_{k}=P_{\bar{k}}H^{T}{\left(HP_{\bar{k}}H^{T}+R\right)}^{-1},$$
(4)$${\hat{x}}_{k}={\hat{x}}_{\bar{k}}+K_{k}(z_{k}-H{\hat{x}}_{\bar{k}}),$$
(5)$$P_{k}=(I-K_{k}H)P_{\bar{k}}A^{T}+Q,$$

Matrix *A* is a status matrix that indicates the relationship between ${\hat{x}}_{k-1}$ and ${\hat{x}}_{\bar{k}}$, *B* is the input control matrix. *Q* is a matrix containing covariance noise and *R* is a matrix containing measurement noise, where both types of noise are independent of each other, are white, and have a normal probability distribution. The matrix *K* is the output of the Kalman filter, which minimizes the resulting covariance error by the given matrix $P_{k}$. The matrix *H* defines the relationship between the current status and measured value.

The practical application of the Kalman filter is shown in Figure 3 and Figure 4. In the first case, filtering was applied to five measured RSSI values, and in the second one, to 350 values. As can be seen in the second figure, Kalman filtration removed the extreme values of signal fluctuation, which is the main benefit. The localization system, however, uses only a few values for determining one RSSI value. In Figure 3, it is possible to demonstrate the main difference between measured data and filtrated values. The result of the average of the raw data is in this case −60.2 dBm, which is at the same time the value of Kalman filtration (dashed). The solid line connects the values measured before the filtration (raw data). However, if the average of the filtered data is calculated, the result is −56.1 dBm, i.e., a value which is not as dependent on the extreme measured data. That is why the designed system utilizes this method of scanning RSSI values.

## 4. Fuzzy System Design

The number of membership functions in the proposed system is determined on the basis of the nature of its input variables, i.e., RSSI values and their difference. Unfortunately, these values are subject to unpredictable fluctuation, which is why it would not be appropriate to divide the entire universe into a large number of intervals. When complex calculations are done, the distinguishing ability of the algorithm diminishes (correct fuzzification), because the variable input values can easily move between adjacent fuzzy sets. On the other hand, it is necessary to guarantee a certain degree of distinguishability between these sets. There are five defined membership functions on the basis of practical experience and the literature (see Figure 5).

As for the shape of the membership function, it can be stated that it affects the accuracy of the system to a lesser extent than the number or position of the function. Due to the fact that the input values are partially affected by signal fluctuation, the difference between the Gaussian function and the triangle-shaped function is more or less removed. The main difference between the function is only a slightly different value of the degree of membership for the same value of the input variable. In terms of localization, it is more important that the value of the variable is fuzzified to the correct fuzzy set. For correct fuzzification, it is also necessary to determine the correct position of the function with respect to the universe and to other functions. The optimization algorithm is used for this purpose; it recalculates the original distribution of the functions into another one. At the beginning, the fuzzy system always has such defined fuzzy sets. The new positions are always calculated so that their value affects (improves) the accuracy of the algorithm as a whole. In a sense, therefore, there is larger differentiation among the results.

### 4.1. Training Set

The training set is used to generate rules and to subsequently optimize them. They are used for the optimization of fuzzy sets. In terms of this work, the proposed fuzzy system is constructed by one input variable, Zone, which expresses the name of the zone where the measurement was performed. It is further defined by five input variables, ΔRss1_Rssi2, ΔRss2_Rssi3, ΔRss3_Rssi4, ΔRss4_Rssi5, and ΔRss5_Rssi6, which represent individual difference values from six access points. Therefore, each record in the database is in the form:

∆Rss1_Rssi2, ∆Rss2_Rssi3, …, ∆Rss5_Rssi6.
(6)

Of course, the name of the zone is also the future result of the localization process. The measurement itself is not dependent on precisely defined measurement points in a given area, so it is possible to measure these data from a static position in a specific zone. It is, therefore, possible to place a measuring probe into a given location to collect the necessary data. This procedure, however, is not recommended, as it limits the set of input values which may occur in the given zone. That is why it is good to compare the given zone in at least five places, creating the shape of a star (center and corners of the zone).

### 4.2. Generating If–then Rules

The second important part of the proposed Fuzzy Rule-Based System (FRBS) is the if–then rules database. There are many methods for designing these rules. As with the design of fuzzy sets, the methodology of expert knowledge is useful here. For more complex models with more input and output variables, it is, however, very difficult to design an appropriate database. That is why methods are used which automatically generate rules from numeric data. Basically, it is training the fuzzy system using training data. In the case of this designed system, these data were measured in a real environment where the localization took place.

The so-called Wang and Mendel (WM) method [23] was chosen for generating the rules because of its fast application, easy interpretation, and good efficiency. On the other hand, it does not achieve the same quality of results in comparison with other methods; this shortcoming can be compensated by using a simulated annealing algorithm.

The procedure for generating the rules can be described as follows:The first step is the creation of the training data set.As all inputs and outputs are known, it is necessary to define their fuzzy sets.Next, it is necessary to create a set of possible rules which can become active rules. In practice, this means that a rule is created inside each record of a training data set according to a general formula in the form of an equation, for example. Here, it is important to note that the value of each input variable is always fuzzified into the respective language of expression according to the proposed membership functions (fuzzy sets).
**if** ∆Rss1_Rssi2 *is low* AND … AND ∆Rss5_Rssi6 *is high***then***zone is zone*1,
(7)Duplicate rules are then removed from the created rule database.Then the so-called degree of importance is calculated for all rules. This is the transformation of the individual degrees of membership (e.g., µ_∆Rss1_Rssi2_ · µ_∆Rss1_Rssi2_ …) from the step of fuzzification into linguistic expressions. This classifies the rules in the sense that the higher the degree of importance, the better the rule is from the system point of view.In the last step, the generated rules are grouped according to the same conditional part of the rule. On the basis of the degree of importance, the rule with the highest value of this degree is chosen from each group. Other rules from the group are removed from the database. This step is important for the elimination of an erroneous derivation of the results, as the fuzzy system would not be able to derive a result from two identical conditions.

### 4.3. If–then Rule Optimization

If–then rule optimization is based on an improved version of the WM method, which includes the Simulated Annealing (SA) algorithm. In the last step, the rules are grouped according to their identical conditional parts and subsequently only the rule with the highest value of importance is kept in the database.

Other rules of the same group are removed, although they could have had a far greater impact on the decision-making of the system as a whole. This means, in particular, the degree of differentiation between the individual results. In other words, the fuzzy system would prefer, with the highest probability, one possible result as opposed to a situation where it could present multiple possible results.

The improved variant of the WM method [24] utilizes simulated annealing [25]. The individual phases of this rule generation are as follows:Steps 1–4 are identical to the previous ones.In the last step, groups are again generated according to identical condition parts. Every group contains several possible results for a given condition. The purpose of the SA algorithm is again the minimization with the help of the Mean Square Error (MSE) function according to relation (8). Now, however, there are not any specific membership function changes, but changes in the parameters of database functions. This means that a database of rules is created according to a number of groups, where each group is defined by a relevant condition. The results (which in this case are the parameters) of the rules, however, change according to what possible results for a given condition in the group are available. The SA algorithm, therefore, looks for such a combination of results for which the MSE function is minimized. After finding the best combination of results and the best database rules, the other unused results are removed.

### 4.4. Optimization of Fuzzy Sets

The optimization of fuzzy sets (their membership function) is a way to improve the accuracy of deriving the result. The result of the design can be optimized using several methods, especially neural networks, genetic algorithms, or, for example, so-called Simulated Annealing (SA). The latter method was used for optimization [26,27].

Simulated annealing is a stochastic optimization method which was developed in the 1980s by two independent authors [28,29]. It is based on the physical principle of metal annealing. This process consists in the slow cooling of a metal so that the minimum energy value of the material is achieved (change of bonds between atoms). This minimum in the SA algorithm then expresses the global minimum of a specific function. It can be any function where the Mean Square Error (MSE) is minimized for the purposes of this work, see Equation (8).
(8)$$MSE=\frac{1}{N}\cdot \sum_{i=1}^{N}{\left({y^{\prime }}_{i}-y_{i}\right)}^{2},$$
where *N* is the amount of data in the training set (the same data that were used to generate the rules) and ${y^{\prime }}_{i}$ is the result of defuzzification based on the input values from the training set (for the case of initial solution S1). Conversely, for the generation of the new solution S2, the input is the changed data compared to the existing solution S1. The initial and current solutions are marked identically, differing only in the moment when they are generated; the initial solution serves only to initialize the current solution.

The term solution means the stored configuration (value) of the parameters of the specific membership function, for which the SA algorithm searches the minimal value of the MSE function. Thus, the initial solution 1 means that, when the SA algorithm is initiated, the input values of the parameters of the membership function are read, see Figure 5. The new solution S2 is generated based on these parameters; if it is better than S1, it is accepted and S1 = S2 is valid, and S1 becomes the current best solution.

The parameter is directly the result of a specific record of the training set (its last part). This minimization is global in nature. This means that the case in which only the local minimum is found does not occur, for example, in a gradient algorithm, which terminates immediately after finding the local minimum/maximum. For this purpose, SA uses the so-called metropolitan acceptance criterion [30], which is based on probabilities. The corresponding Equation (9) expresses the probability that the SA algorithm will accept the rejected solution from the first phase, thus allowing the algorithm to overcome local extremes.
(9)$$p(T)=e^{-\Delta E/T},$$

The goal of this optimization is to find new positions of membership functions for the given input variable and the given fuzzy set. The resulting optimized positions should provide better result derivation with reference to the fuzzy system as a whole.

Unlike rule optimization, fuzzy set optimization can be clearly displayed and compared. Figure 6 shows the original design of the fuzzy set for input variables. It is the initial design for all generated fuzzy systems; this means that the result of the optimization is dependent only on the training data. The possible result is displayed in Figure 7, where the displacement of some fuzzy sets can be clearly seen. In this part of the optimization, only the displacement parameters are optimized; of course, it is possible to optimize the width or the position of the individual membership functions. The graphs were designed with the help of the Java library jFuzzyLogic. The shape of each output fuzzy set is modified to define one zone, see Figure 8.

## 5. Proposed Localization System

### 5.1. System Functionality

The designed localization system consists of a number of elements, which are diametrically different from each other. For easier identification, the name of each individual system element always starts with the key expression FuzzyLoc. The entire localization system is called the *FuzzyLoc System*. The key system requirements include:The creation of a mobile application *FuzzyLoc Admin* for the system administrator, which will be able to train a data set, as well as register the users and access points to the system.The creation of a mobile application *FuzzyLoc Client*, which will scan the surrounding RSSIs; it will be able determine its position (zone) and then send the result to the server on the basis of the downloaded definition of the fuzzy derivation system.The creation of a simple web interface *FuzzyLoc Supervisor* meant for displaying the results of localization and zone definitions, according to which it will identify the resulting zone.The creation of the web server *FuzzyLoc Server*, which will be able to communicate with the abovementioned elements; it will contain a MySQL database for storing all necessary data; its deployment in the production environment will be simple.The creation of the program *FuzzyLoc Engine* for generating definitions, according to which each device will be localized.

Before initiating the localization, it is necessary to set up the server in the local network. Virtual tools solutions are used for this purpose; it is easy to import files into the virtual environment, e.g., VMware. It uses a standardized universal format Open Virtualization Format (OVF), so it is almost certain that it will be compatible with any specific virtual environment. If it is not possible to directly import it, it can be converted to a different file type.

The entire functioning of the system is shown in Figure 9. The individual steps are described in the text below; they are marked with a corresponding step number in the diagram.
The system configuration starts with step 1; a map of the area, where localization will take place, is uploaded through a web interface. The map is uploaded directly onto the server; it is accessible to all devices that will access the surveillance web environment. The maximum allowed map size is 8 MB.The next step is the creation of zones on the basis of the uploaded map. The individual zones are created by defining two corners (upper left and bottom right) of the zone; it is only possible to create zones in the shape of a rectangle. Each zone must be given a unique name according to the needs of the administrator. These zone parameters are then saved on the server.The created zone definitions are then downloaded to FuzzyLoc Admin; on their basis, the training data set is created. Of course, measurements must be done in such a way that the correct zone is chosen from the list, the one where the training data are currently being measured. The data file is then uploaded to the server.The administrator then has to register users (to be localized) and access points (to be used in the localization process —at least six of them) from the mobile application.The administrator then initiates the prepared script/home/user/FuzzyLocEngine/generateFis.sh and, depending on the amount of data, it creates the file fis.fcl.Then the administrator must generate certificates for individual devices to be localized. This step is currently not available due to an error in a specific library in Android OS, so it is temporarily necessary to install only the root certificate FuzzyLoc CA into the client’s device or to utilize one’s own certification resources. The root certificate also needs to be installed onto the PC from which the FuzzyLoc Supervisor will be accessed.Now the localization of the device may start. When a user runs FuzzyLoc for the first time, he/she is prompted to consent to the downloading of fis.fcl from the server. The application will also ask for the SSID and IP server. The user will then only run the localization of the device and the application will switch to background mode. It will send the localization results to the server.The final step is to enter the URL of the https://IP-server on the PC with the root certificate installed. After signing in, the monitoring environment is displayed.

### 5.2. Key System Components

#### 5.2.1. FuzzyLoc Server

As was mentioned before, the designed localization system is a centralized solution, which uses the server to its advantage. It should be noted that the calculation of the mobile devices’ positions is made independently for each of them. The main purpose of the server is to connect all parts of the system into one integrated environment. The main task of the server requirement is communication with mobile applications, i.e., *FuzzyLoc Admin* and *FuzzyLoc Client*. Then, it is necessary to store the system configuration data—for this, a MySQL database is used. The server is also important for accessing results of the localization with the help of the monitoring environment *FuzzyLoc Supervisor*. The last function is the generation of the derivation system with the help of the Java application *FuzzyLoc Engine*.

#### 5.2.2. FuzzyLoc Client

The first application for the Android platform is *FuzzyLoc Client*. The primary task of the application is to calculate the position of the device and subsequently to submit the result to the central server. The application is therefore intended for a user whose position should be identified. For this application and the following one (*FuzzyLoc Admin*), the minimum supported version of the operation system is 4.0.3 (“Ice Cream Sandwich”). For the purposes of localization, the application must have a user environment; however, it is not necessary to display the current position on the user device—the interface solely performs a configuration function, meaning it serves only to set network parameters (server IP, network SSID, etc.).

#### 5.2.3. FuzzyLoc Admin

The second mobile application is *FuzzyLoc Admin*. Unlike the previous application, it is used for system administration. It is therefore used by the administrator, not by the user. One of the requirements is the ability to register new users and access points to the database. A much more complex task for this application is to provide an interface for the creation of training data sets.

#### 5.2.4. FuzzyLoc Supervisor

The last element of the localization system is the *FuzzyLoc Supervisor*, which is a web interface for monitoring and administration. In terms of administration, there are two basic steps. The first step is the uploading of a map onto the server in the form of a picture. This picture then serves to subsequently graphically display individual user positions. It is also used for the second step of the administration, which defining the respective zones. The zones (their number and distribution) are set by the administrator. The zones can be of any size, i.e., one or more rooms, or even an entire building block; it all depends on the localization requirements. The creation of the zone is done by two clicks (taps) on the map, which define the upper-left and bottom-right corners of the zone; the result is a rectangle. After making this definition, the administrator is prompted to set the name of the zone; the name should of course characterize the purpose of the zone. For better distinction, the zones can have different colors. Finally, it is necessary to store all the information on the server by pressing the dedicated button.

The second purpose of the interface is to display the users’ positions on the uploaded map. The map can be zoomed in or out (including in the previous case). There are two different display modes. The first one is the overall view, which marks each zone with a tag and a number, giving the number of users/devices in it. If it is necessary to find out who is located in the given zone, it is possible to move the cursor of the mouse over the tag—a pop-up information window appears, showing a list of present users. To find a particular user more quickly, there is a function where the user can be clicked on in a table; the zone where the user is located is then highlighted. The second display mode is individual. This means that three specific users picked from the table will be displayed on the map individually with a detailed description. As for the table itself, every record here is defined by its name, whether it is a device or a user, and by the name of the zone. In the individual mode, a graphic tag appears to show whether the user is selected to be displayed on the map.

## 6. Methodology and Performed Testing

The designed localization algorithm was subjected to two tests, each taking place in a different environment. The methodology of the test was the same for both environments. Firstly, six access points were deployed (with regard to the used Wi-Fi channels). The environment was then divided into zones, which basically copied the individual rooms. Such a division, however, is not necessary; several rooms can be defined as one zone, or one room (e.g., corridor) can be divided into several zones.

In each zone, there were several defined points at which the *FuzzyLoc Admin* application collected training data. Two measurements took place at each point. The entire training set was then entered into the *FuzzyLoc Engine*, which generated the description of the derivation fuzzy system into the fis.fcl file. It was then entered into the application *FuzzyLoc Client*, which for testing purposes did not send data to the server, but directly displayed the result. The testing took place in the defined points and the assistant held the mobile phone in a specific direction. At each point, 20 position results were generated; zones where the system assumed the location of the device. The resulting accuracy evaluation was therefore given by the successful identification of the correct zone.

To verify this concept, only static tests were performed in the first step. The main objective of this testing was to improve the quality of the developed localization system by finding errors in the early stages of the development cycle and prepare the core system for future development, leading to practical deployment.

### 6.1. The First Test Environment

The first environment is displayed in Figure 10. Six zones were defined on the basis of the available rooms. The corridor was divided into two zones, just like area 304. The environment here was very diverse in terms of the equipment, furniture, open spaces, etc. It was also a very difficult environment to perform localization, because there were many factors affecting the final resulting success. The first factor was the movement of people, as the measurement was carried out during working hours. This movement of people includes the opening and closing of doors and the training data were measured with closed doors, but due to external factors, the doors were open during localization estimation. This concerns in particular zones 1, 3, and 5. Aside from this, the measurement by RFID technology was in progress at the time, so even this could have affected the accuracy of the results. The final factor, which is more a characteristic of the environment, is the imaginary triangle created by the localization points (rectangular with an arrow) (5), (6), and (8). There were no classic doors leading into zone 2; the triangle shape can therefore be considered as one area, which explains the results.

### 6.2. The Second Test Environment

The diagram of testing in the second environment is shown in Figure 11. The testing procedure was identical to the previous one. There were seven zones defined in the area, which were differentiated by color, and the corridor was again divided into two smaller zones. The measurement took place in the afternoon, therefore in a less busy environment. The environment was, as opposed to the previous one, more structured and bigger and corresponded to a typical model of a corridor leading into separated offices on each side. For room-level localization, it was an ideal environment; each room was far more distinguishable from the others thanks to a specific attenuation of the signal in the individual rooms. The signal was affected by the multiple divisions of the area, but also by the thickness of the walls or their material. It can also be affected by security doors, which were not present in the previous environment. It is important to note that the previous testing environment was rather open, and that is why the success rate of the localization was low.

At first glance, it may seem that an AP was placed in each zone, which may lead to the thought that this configuration (1 zone = 1 AP) is necessary for localization. This assumption could have been valid for the previous version of the localization system (see [22]) where the highest level signal was taken into account. Therefore, the highest level could have the highest weight in the calculation itself. In the present system, the method for the scanning of differential values *ΔRSSI* between the specific access points was utilized, and so all were equal to each other and all affected the successfulness in the same way. This is demonstrated by localization points (9) and (10), which were successful despite the absence of APs in the given zone. The individual localization points and their corresponding success rates are shown in Table 3. There is an evident difference in the success rates in both tests, which proves that the localization system is strongly dependent on the specific environment. In practice, this means that it is suitable for divided spaces such as hospital environments, but not so much for modern open office spaces.

### 6.3. Results and Discussion

The developed localization system was tested in two different environments. The first one was a rather open learning environment, while and the second one was a standard building floor with a number of rooms. Table 3 compares the success rates of the individual tests. From these results, it can be concluded that, in a standard environment where rooms are normally separated by partitions and doors, the localization system is able to derive the position of the device far better than in an open environment. For this reason, the individual rooms are unique from the perspective of signal coverage. That is why there is a high level of distinguishability between the individual zones, greater than in a more open environment.

From Table 3, it can be concluded that the localization system tended to localize with better success in zones which are, in effect, standard rooms. This was true for localization points (3), (4), (7), (9), and (10). Here, the distinguishability of rooms was at a good level. Localization point (2) was affected the most by the mentioned factors, and that is why it cannot be considered a reliable measure for assessing the success of the system. The error rate in the mentioned triangle shape was mainly caused by the low distinguishability between zones. This means that, if the zones were separated by partitions, each zone would have its characteristic coverage map and the system would be able to detect them better.

The determination of the resulting accuracy of the previous localization system was performed using a series of repeated measurements, where the individual points of the database from the training phase were created with a one-meter distance. The access points were set to a constant transmitting power. As for the abovementioned factors, multipath signal propagation through closed doors was partially limited and one type of mobile device was used. The measurement itself took place in four stages, each containing 40 position calculation intervals.

The resulting localization accuracy is shown in Table 4. The original system—the Weighted k Nearest Neighbors (WkNN) method (see [21])—thus achieved a 78.1% success rate in determining the correct room where a user is located. The average distance error was 2.65 m. Subsequently, tests of the method based on fuzzy logic were performed, while other rooms on the floor were taken into account within the testing. In the case of comparing the same localization success rate at the level of the same rooms, the optimized system based on fuzzy logic achieved a success rate of 90%; in the case of including other rooms in which the original measurement was not performed, the success rate was 85.9%. This is mainly due to the low accuracy at the level of point (11), where the low value was given mainly by the position of the nearest access point, which was located in a room that did not provide ideal conditions for signal propagation.

The following basic geometry equation was used to calculate the distance error, where *x*, *y* is the actual position and $\bar{x}$, $\bar{y}$ is the position determined by the system.
(10)$$E=\sqrt{{\left(x-\bar{x}\right)}^{2}+{\left(y-\bar{y}\right)}^{2}},\;$$

As can be seen in Figure 12, the zone is defined as a room. For zone creation, the zone administration interface was *FuzzyLoc Supervisor*. As mentioned above, the main criterion of the localization system is its accuracy, which depends on the so-called degree of localization level. In this concept, we tried to develop a localization system at the zone level. These are places where there are often a large number of people or medical devices. In the case of managing crisis situations, it is sufficient to have an overview of, for example, how many people are currently in a given zone, etc. A more precise type of localization is not usually required in desired cases.

As mentioned above, extensive preliminary measurements had been made using various equipment (see the example in Figure 13) before the system implementation began, and some refinement measurements were also necessary during the implementation itself. Common Wi-Fi features had to be taken into account when designing the software part.

The developed system relies on the fingerprint database that has to be defined using a simple graphical interface (see Figure 14) and subsequently stored on a network server—therefore, the current version of the database has to be downloaded to the client device before the localization starts, along with the map (image) of the area.

Then, the client application scans for the available access points, evaluates the signal levels and displays the current position in the map. The look of the user interface is very similar to the survey mode (Figure 14), just the red cross is missing and the current location is marked with a single green point in the map.

## 7. Conclusions

The article deals with the design and optimization of a system for the localization of mobile devices inside buildings on a zone level. A zone can be defined as a room, multiple rooms, or a part of a building. The carrier medium is the local wireless network (Wi-Fi technology). There are a number of factors associated with using this technology which diminish the successfulness of localization. The two most important ones are signal fluctuation and the variation of Wi-Fi adapters. Neither of them can be entirely eliminated, that is why methods were designed which could at least minimize their effect. Signal fluctuation is minimized by the Kalman filter; its main characteristic is resilience against extreme measured RSSI values (measured signal level). The localization system utilizes RSSI values as a key metric for deriving device positions. For this reason, these values for individual measurements are averaged from several samples. However, if directly measured data are averaged, there is a high probability that the result will be too affected by an extreme value present among them (extremely low or extremely high signal level). These extremes, of course, affect the resulting average. The Kalman filter does not filter out the extreme values completely, on the contrary, it takes them into account, but as opposed to standard averaging, the resulting values are not so much affected by the extremes.

The second key problem consists in the differences among Wi-Fi adapters. This concerns the adapters of mobile devices by various manufacturers, but also devices of the same manufacturer. There are also differences between two identical devices which could have the same type of adapter or a different version of the operating system. The differences between the adapters result in different measured RSSI values, although they are measured at the same place, at the same time, with the same orientation of the two devices. The range of various models is so wide that it is not possible to create a database containing the relevant properties of all compatible devices. Even if it is created, there would be an influencing factor, and that is the OS version. That is why a method was designed in which the absolute values from individual access points are not measured, but rather their differential values, that is, *Δ_AP1-AP2_*, *Δ_AP2-AP3_*, etc. The differential value should theoretically be the same even for different adapters.

The main purpose of this paper was to compare two approaches, where, for the second one, fuzzy logic was used to refine the localization based on the fingerprinting method. Not only people, but also hospital devices and equipment located in the rooms (zones) can be considered as objects to localize. For this reason, the localization was performed at the zone level, where a zone was defined as a room. In the first phase, only static tests were performed. As part of the future development of this concept, we plan to refine our solution by not only verifying the new data set and the test data, but we also plan to perform dynamic tests. Future work should instead be on the influence of the application of current fuzzy logic on accuracy and devoted to a comparison of the results of localization using the fingerprinting method and evaluating the differences between static and dynamic localization tests. This complex task will require a number of modifications in process methodology, as well as data processing changes such as different configuration settings of the Kalman filter, etc.

## Figures and Tables

**Figure 1 sensors-20-04490-f001:**
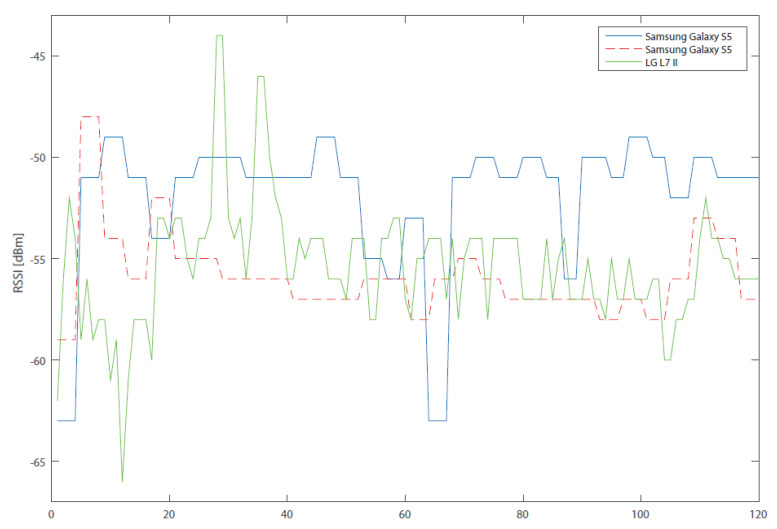
Comparison of Received Signal Strength Indication (RSSI) measurements for multiple devices.

**Figure 2 sensors-20-04490-f002:**
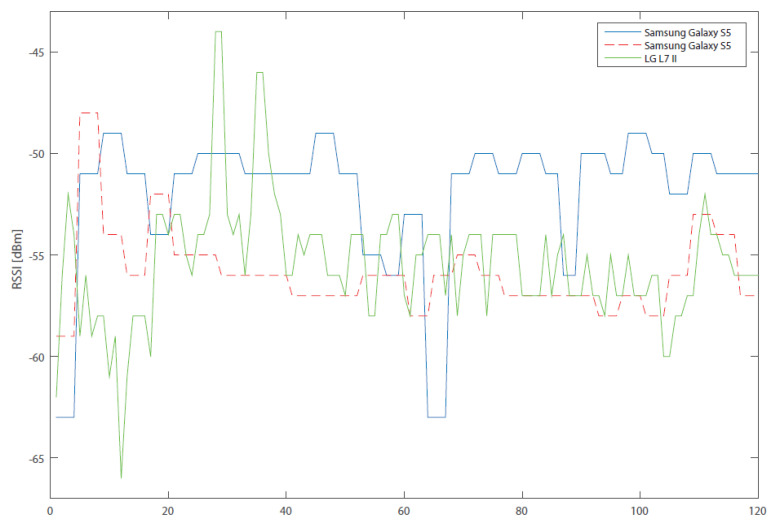
Comparison of RSSI measurements for multiple devices (Kalman filter).

**Figure 3 sensors-20-04490-f003:**
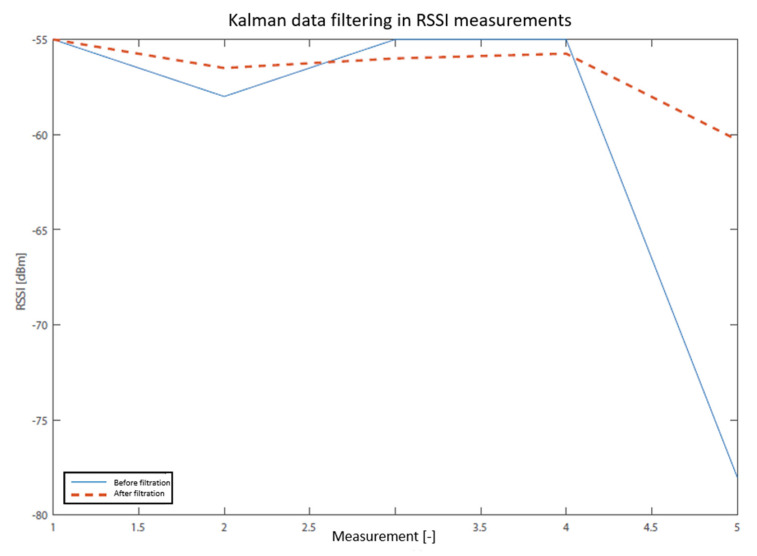
RSSI data filtering by Kalman filter.

**Figure 4 sensors-20-04490-f004:**
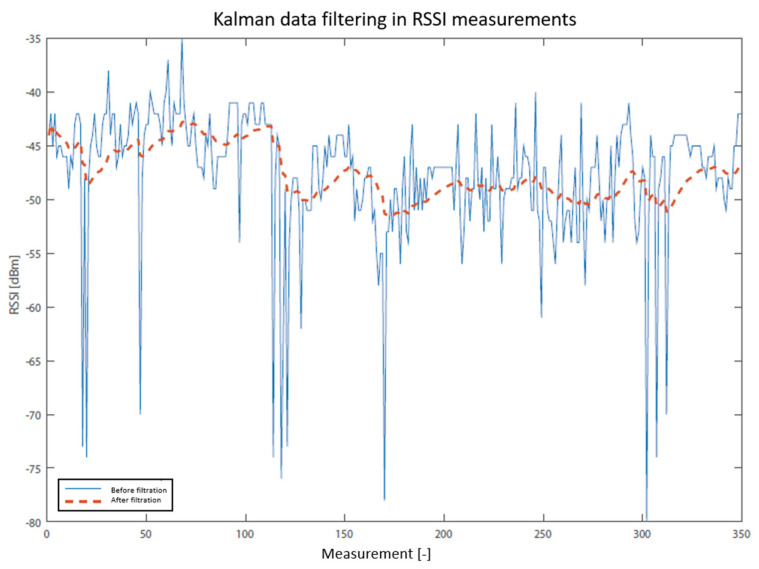
RSSI data filtering by Kalman filter.

**Figure 5 sensors-20-04490-f005:**
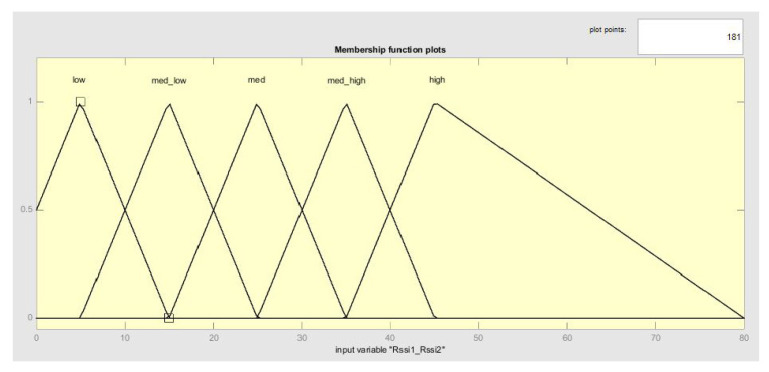
Proposed fuzzy sets for input variable ΔRss1_Rssi2.

**Figure 6 sensors-20-04490-f006:**
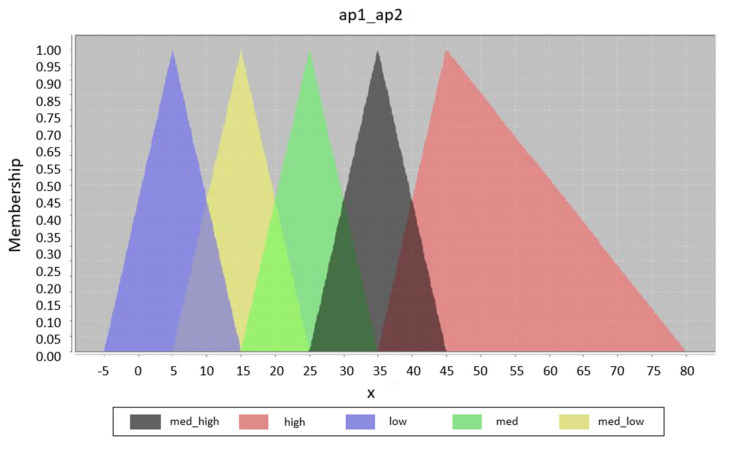
Initial fuzzy set design.

**Figure 7 sensors-20-04490-f007:**
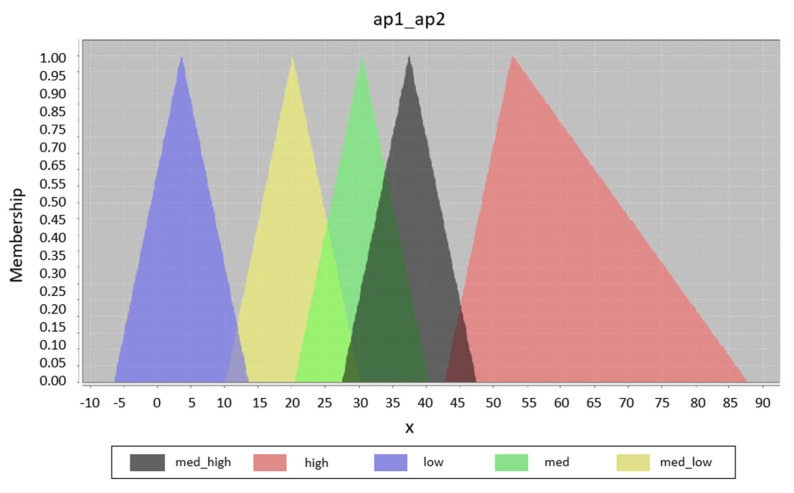
Optimized fuzzy sets for initial design.

**Figure 8 sensors-20-04490-f008:**
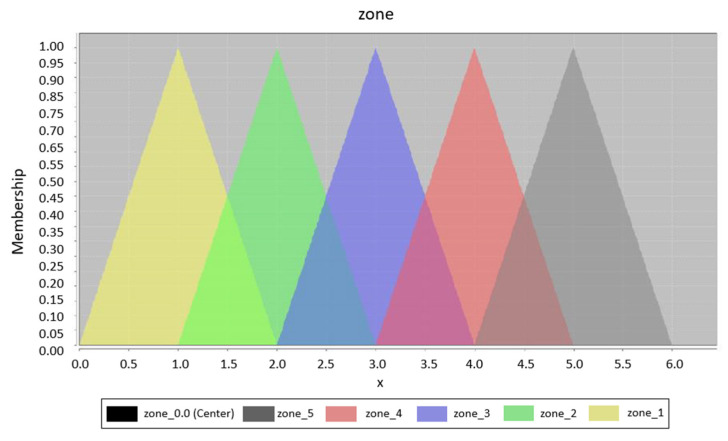
Output fuzzy sets.

**Figure 9 sensors-20-04490-f009:**
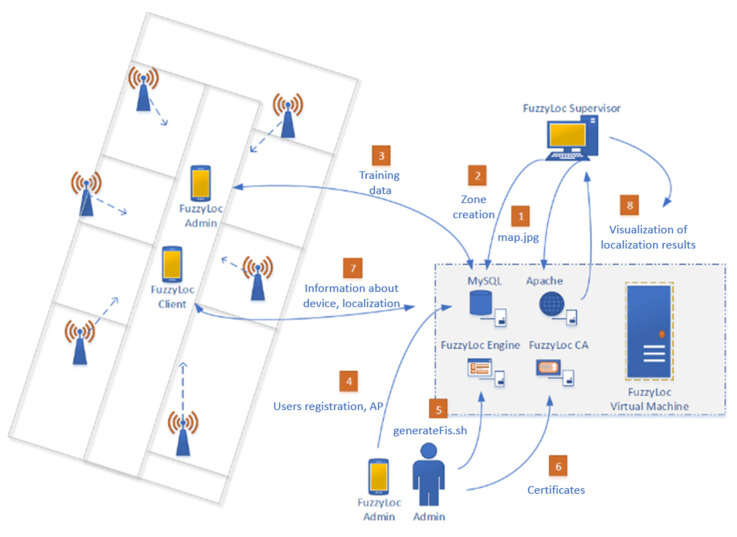
Localization system diagram.

**Figure 10 sensors-20-04490-f010:**
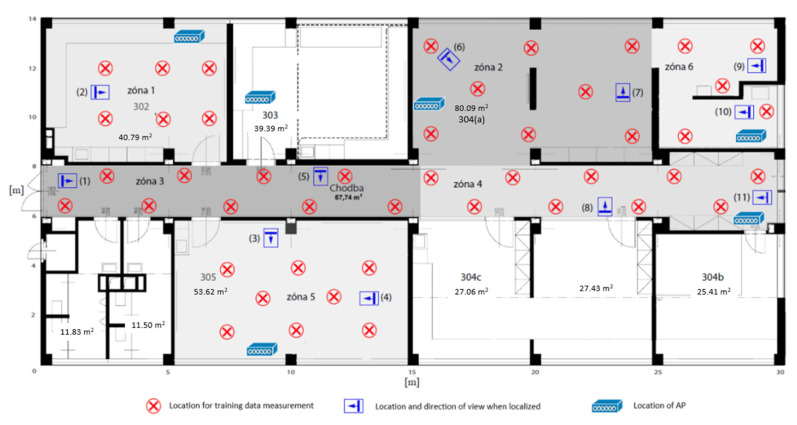
Test environment 1.

**Figure 11 sensors-20-04490-f011:**
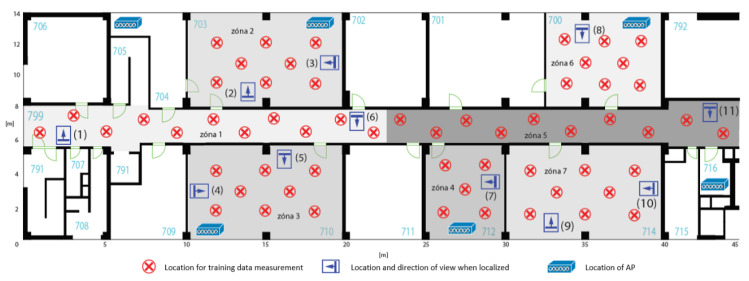
Testing environment 2.

**Figure 12 sensors-20-04490-f012:**
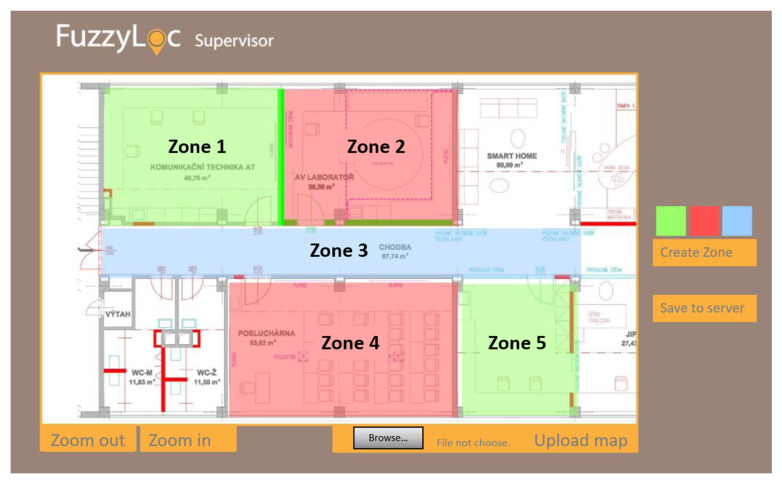
FuzzyLoc Supervisor, zone administration interface.

**Figure 13 sensors-20-04490-f013:**
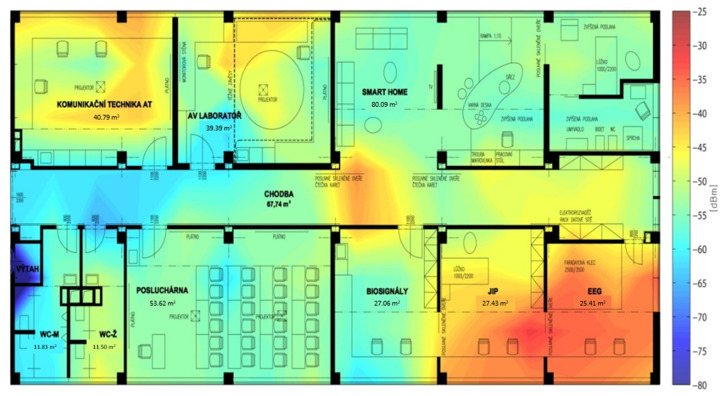
Visualization of signal level in the considered area—an example.

**Figure 14 sensors-20-04490-f014:**
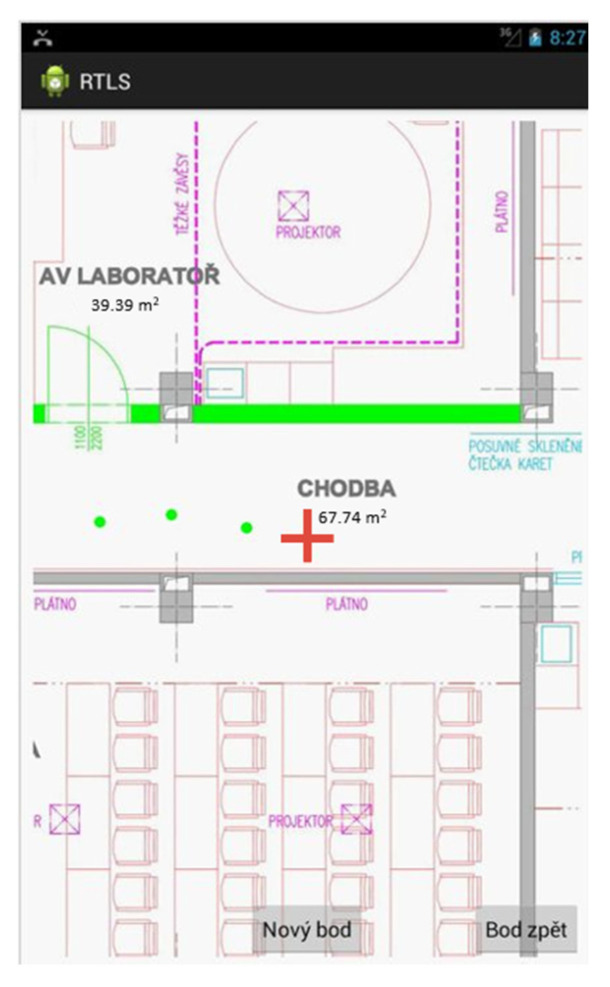
Survey mode of the client application—database creation.

**Table 1 sensors-20-04490-t001:** Comparison of parameters of technologies for internal localization [2].

Technology	Network Throughput	Frequency Band	Range	Security
Bluetooth	2 Mbps	2.4 GHz	800 m	255bit AES
RFID (NFC)	106–424 kbps	13.56 MHz	20 cm	TIP
RFID (UHF)	40 kbps	860–960 MHz	12 m	none
Wi-Fi	1–300 Mbps	2.4 and 5 GHz	50 m	SSID
IrDA	14.4 kbps	850–900 nm	0–1 m	IRFM
UWB	53–480 Mbps	3.1–10.6 GHz	0–10 m	High
ZigBee	20–250 kbps	868 Mbps–2.4 Gbps	10–75 m	128bit AES

**Table 2 sensors-20-04490-t002:** Comparison of different localization methods with regard to the accuracy and deployment complexity.

RTLS Technology	Gu et al. [5]
RFID	2–3 m
GPS	15 m
UWB	0.15 m
Visual analysis	-
WLAN	4 m (2D)
Ultrasonic	0.03 m
IR	3 mm

**Table 3 sensors-20-04490-t003:** Successful determination of the correct zone in each environment (%).

Positioning Point No.	Testing Environment 1	Testing Environment 2
1	65	100
2	40	100
3	75	90
4	80	95
5	65	95
6	50	80
7	75	85
8	50	90
9	80	90
10	90	75
11	55	45

**Table 4 sensors-20-04490-t004:** Comparison of successful determination of the correct zone and average distance error for testing environment 2.

Positioning Point No.	“WkNN” Localization		Fuzzy Logic-Based Localization	
	Success Rate [%]	Average Distance Error [m]	Success Rate [%]	Average Distance Error [m]
1	-	-	100	2.12
2	-	-	100	2.03
3	80	2.77	90	2.42
4	-	-	95	2.17
5	70	2.49	95	2.19
6	-	-	80	2.61
7	82.5	2.11	85	2.41
8	80	3.24	90	2.34
9	-	-	90	2.77
10	-	-	75	2.82
11	-	-	45	2.86
	78.1	2.65	85.9	2.43

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
