# Peer review of "Indoor Positioning System Based on Fuzzy Logic and WLAN Infrastructure†"

_sensors, 2020, doi:10.3390/s20164490_

Round 1

Reviewer 1 Report

The authors present an interesting paper about indoor localisation, in wich they make a merge of several approaches/methods to obtain better results than those obtained by more traditional approaches. However there are some issue that deserve some improvements, namely:

  1. a review of the English writing is recommended as it presents some typos: "localization is system is"; "have various accuracy"
  2. The introduction section should be more supported with bibliographic references
  3. while the part referring to the description of the method proposed by the authors and its stages is well described, the same is not true of the test conditions, which deserved better and more detailed information. Also more numerical examples (eg tables), would be more illuminating of the effects of improvements that the proposed method allows to obtain, when compared with other methods. The authors almost summarize this part of the paper to the description of the apps and features / function from a macro point of view, in view of a deeper analysis of the effect of these approaches on the data obtained.
  4. During the article, the authors describe the qualities of their method, which seemed to be applied in determining the location (point), but in the end the classification used is zonal. What is the cost / benefit complexity / accuracy?
  5. Still in relation with the results it would be interesting to have an explanation of the cases in which the "classification of the location is poorer"
  6. The conclusions should include a more quantitative analysis, that is, how much better is this system when compared to other works? How much better is the system? what restrictions does the system have for the results to be, for example, greater than 85% or 90%?

Finally, congratulations on your work

Author Response

Dear reviewer,

Thank you so much for your valuable comments and recommendations. We really appreciate time you spent to make a review of our paper. Based on your remarks we have tried to do our best to enrich our article as you suggested. The changes are indicated in an updated version by green color. We really hope that these modifications and addition of some parts will contribute to accept the paper.

1. A review of the English writing is recommended as it presents some typos: "localization is system is"; "have various accuracy"

Thank you so much for your remarks. The article was read again independently by the author's team and various errors and typos were removed. Subsequently, the article was subjected to proof-reading by an expert through a translation agency.

2. The introduction section should be more supported with bibliographic references. While the part referring to the description of the method proposed by the authors and its stages is well described, the same is not true of the test conditions, which deserved better and more detailed information. Also more numerical examples (eg tables), would be more illuminating of the effects of improvements that the proposed method allows to obtain, when compared with other methods. The authors almost summarize this part of the paper to the description of the apps and features / function from a macro point of view, in view of a deeper analysis of the effect of these approaches on the data obtained.

Thank you for this comment. We add new section called related work where we have tried to summarize the state of the art regarding Indoor Positioning Systems and Localization techniques. Based on your recommendations we also did a comparison of these techniques and approaches and listed the main pros and cons.

3. During the article, the authors describe the qualities of their method, which seemed to be applied in determining the location (point), but in the end the classification used is zonal. What is the cost / benefit complexity / accuracy?

Thank you so much for this comment. It may be a little bit confusing. Based on our research, the selection of the right technology might not be an easy task since many features have to be balanced, such as target application, deployment costs, required accuracy, tolerable uncertainty or needed computational resources. In general, each base indoor positioning technology has a well-defined domain of applications: Wi-Fi fingerprinting is usually applied in smartphone applications, whereas UWB is more suitable for complex applications where higher accuracy is required.

For our purposes - localization of mobile devices in the building environment, suitable for use in health care or crisis management - is absolutely satisfactory to determine the position of a client device on the level of rooms, i.e. zone. This is the main reason why we dealt with “zone localization”. The other benefits are cost, this approach is not time consuming and is labor effective.

4. Still in relation with the results it would be interesting to have an explanation of the cases in which the "classification of the location is poorer"

Thank you so much for your remarks. We have tried to explain and describe the test scenarios in more detail.

5. The conclusions should include a more quantitative analysis, that is, how much better is this system when compared to other works? How much better is the system? what restrictions does the system have for the results to be, for example, greater than 85% or 90%?

Thank you for this recommendation. We have tried to point out the main benefits of our approach. We add some comments to support our solution to the results and discussion section.

We would like to thank you again and apologize for your time you spent with reviewing of the revised version of our manuscript. Thank you for your contribution.

Reviewer 2 Report

In this paper, the authors propose an indoor positioning system (IPS) based on Wi-Fi technology, received signal strength (RSS), and fuzzy logic to provide room-level localization. The main focus of the authors' work is on the signal fluctuation and heterogeneity of Wi-Fi adapters. Although the work is pleasing, up-to-date, and the proposed framework very interesting, the authors lack proper algorithm evaluation and contextualization of the work. For those reasons, in the current form, I do not recommend the publication of the work. My comments and suggestions are as follows:

  1. the authors must improve the paper readability. In the current version, it is complicated to follow the ideas on the paper as the language is not clear. Proofreading is mandatory;
  2. Although the literature of IPSs is plentiful, especially for fingerprinting-based ones, the authors completely missed a proper literature review. There are several sentences that require a proper reference to support them. The authors must describe similar methods in the literature and compare them with the proposed one. In the current version, it is not clear why their method is superior. Some suggestions of references that must be included, but not limited to, are:
    • S. He and S.-H. G. Chan, “Wi-Fi Fingerprint-Based Indoor Positioning: Recent Advances and Comparisons,” IEEE Commun. Surv. Tutorials, vol. 18, no. 1, pp. 466–490, Jan. 2016. 
    • Fuchs, N. Aschenbruck, P. Martini, and M. Wieneke, “Indoor tracking for mission critical scenarios: A survey,” Pervasive Mob. Comput., vol. 7, no. 1, pp. 1–15, Feb. 2011.
    • C. Fischer and H. Gellersen, “Location and Navigation Support for Emergency Responders: A Survey,” IEEE Pervasive Comput., vol. 9, no. 1, pp. 38–47, Jan. 2010.

      A. F. G. Ferreira, D. M. A. Fernandes, A. P. Catarino, and J. L. Monteiro, “Localization and Positioning Systems for Emergency Responders: A Survey,” IEEE Commun. Surv. Tutorials, vol. 19, no. 4, pp. 2836–2870, 2017.

    • S. Yang, P. Dessai, M. Verma and M. Gerla, "FreeLoc: Calibration-free crowdsourced indoor localization", Proc. IEEE INFOCOM, pp. 2481-2489, 2013.
    • A. Farshad, J. Li, M. K. Marina and F. J. Garcia, "A microscopic look at WiFi fingerprinting for indoor mobile phone localization in diverse environments", Proc. IPIN, pp. 1-10, 2013.
    • C. Wu, Z. Yang, Y. Liu and W. Xi, "WILL: Wireless indoor localization without site survey", IEEE Trans. Parallel Distrib. Syst., vol. 24, no. 4, pp. 839-848, Mar. 2013.
    • A. Ferreira, D. Fernandes, A. Catarino, and J. Monteiro, “Performance Analysis of ToA-Based Positioning Algorithms for Static and Dynamic Targets with Low Ranging Measurements,” Sensors, vol. 17, no. 8, p. 1915, Aug. 2017.
    • Building a practical Wi-Fi-based indoor navigation system", IEEE Pervasive Comput., vol. 13, no. 2, pp. 72-79, Apr. 2014.
    • V. Honkavirta, T. Perala, S. Ali-Loytty and R. Piché, "A comparative survey of WLAN location fingerprinting methods", Proc. IEEE WPNC, pp. 243-251, 2009.
    • A. G. Ferreira, D. Fernandes, A. P. Catarino, A. M. Rocha, and J. L. Monteiro, “A Loose-Coupled Fusion of Inertial and UWB Assisted by a Decision-Making Algorithm for Localization of Emergency Responders,” Electronics, vol. 8, no. 12, p. 1463, Dec. 2019.
    • B. Lu, J. Niu, J. Juny, L. Cheng and Y. Guy, "WiFi fingerprint localization in open space", Proc. IEEE LCN, pp. 1-4, 2013.
  3. The authors must provide a proper evaluation of the methods proposed (variety of Wi-Fi adapter and signal fluctuation). At least, the authors must present the results with and without those improvements to validate the performance improvement with their use;
  4. The fuzzy rules are optimized only on the training set? Are they tested on different, unseen, data to validate that no overfitting is being made?
  5. How many tests per point are performed?
  6. are the tests dynamic? What is the accuracy with dynamic tests vs static ones?
  7. in the abstract, the authors state that the accuracy of their system is 2.5m but, on the evaluation part there is no information that supports that sentence.

Author Response

Dear reviewer,

Thank you so much for your valuable comments and recommendations. We really appreciate time you spent to make a review of our paper. Based on your remarks we have tried to do our best to enrich our article as you suggested. The changes are indicated in an updated version by green color. We really hope that these modifications and addition of some parts will contribute to accept the paper.

1. The authors must improve the paper readability. In the current version, it is complicated to follow the ideas on the paper as the language is not clear. Proofreading is mandatory;

Thank you so much for your remarks. The article was read a few more times independently by the author's team and various errors and typos were removed. Subsequently, the article was subjected to proof-reading by an expert through a translation agency.

2. Although the literature of IPSs is plentiful, especially for fingerprinting-based ones, the authors completely missed a proper literature review. There are several sentences that require a proper reference to support them. The authors must describe similar methods in the literature and compare them with the proposed one. In the current version, it is not clear why their method is superior. Some suggestions of references that must be included, but not limited to, are:

  • S. He and S.-H. G. Chan, “Wi-Fi Fingerprint-Based Indoor Positioning: Recent Advances and Comparisons,” IEEE Commun. Surv. Tutorials, vol. 18, no. 1, pp. 466–490, Jan. 2016. 
  • Fuchs, N. Aschenbruck, P. Martini, and M. Wieneke, “Indoor tracking for mission critical scenarios: A survey,” Pervasive Mob. Comput., vol. 7, no. 1, pp. 1–15, Feb. 2011.
  • C. Fischer and H. Gellersen, “Location and Navigation Support for Emergency Responders: A Survey,” IEEE Pervasive Comput., vol. 9, no. 1, pp. 38–47, Jan. 2010.
  • F. G. Ferreira, D. M. A. Fernandes, A. P. Catarino, and J. L. Monteiro, “Localization and Positioning Systems for Emergency Responders: A Survey,” IEEE Commun. Surv. Tutorials, vol. 19, no. 4, pp. 2836–2870, 2017.
  • S. Yang, P. Dessai, M. Verma and M. Gerla, "FreeLoc: Calibration-free crowdsourced indoor localization", Proc. IEEE INFOCOM, pp. 2481-2489, 2013.
  • A. Farshad, J. Li, M. K. Marina and F. J. Garcia, "A microscopic look at WiFi fingerprinting for indoor mobile phone localization in diverse environments", Proc. IPIN, pp. 1-10, 2013.
  • C. Wu, Z. Yang, Y. Liu and W. Xi, "WILL: Wireless indoor localization without site survey", IEEE Trans. Parallel Distrib. Syst., vol. 24, no. 4, pp. 839-848, Mar. 2013.
  • A. Ferreira, D. Fernandes, A. Catarino, and J. Monteiro, “Performance Analysis of ToA-Based Positioning Algorithms for Static and Dynamic Targets with Low Ranging Measurements,” Sensors, vol. 17, no. 8, p. 1915, Aug. 2017.
  • Building a practical Wi-Fi-based indoor navigation system", IEEE Pervasive Comput., vol. 13, no. 2, pp. 72-79, Apr. 2014.
  • V. Honkavirta, T. Perala, S. Ali-Loytty and R. Piché, "A comparative survey of WLAN location fingerprinting methods", Proc. IEEE WPNC, pp. 243-251, 2009.
  • A. G. Ferreira, D. Fernandes, A. P. Catarino, A. M. Rocha, and J. L. Monteiro, “A Loose-Coupled Fusion of Inertial and UWB Assisted by a Decision-Making Algorithm for Localization of Emergency Responders,” Electronics, vol. 8, no. 12, p. 1463, Dec. 2019.
  • B. Lu, J. Niu, J. Juny, L. Cheng and Y. Guy, "WiFi fingerprint localization in open space", Proc. IEEE LCN, pp. 1-4, 2013.

Thank you for this comment. We really appreciate the time you spent to provide us a really useful references. We carefully studied that literature and we add new section called related work where we have tried to summarize the state of the art regarding Indoor Positioning Systems and Localization techniques. Based on your recommendations we also did a comparison of these techniques and approaches and listed the main pros and cons.

3. The authors must provide a proper evaluation of the methods proposed (variety of Wi-Fi adapter and signal fluctuation). At least, the authors must present the results with and without those improvements to validate the performance improvement with their use;

Thank you so much for your remark. Only the variety of Wi-Fi adapter elimination setup was used (utilizing the RSSI difference method), the problem with using raw RSSI values is quite well known. To minimalize the effect of signal fluctuation the Kalman filter has been used, while the results can be seen in Fig.3 and Fig 4.

4. The fuzzy rules are optimized only on the training set? Are they tested on different, unseen, data to validate that no overfitting is being made? Indeed.

Thank you so much for your comment. The fuzzy rules were optimized on the same data that were used for rules generation.

5. How many tests per point are performed?

Thank you so much for this comment. The determination of the resulting accuracy of the previous localization system was performed using a series of repeated measurements, where the individual points of the database from the training phase were created with a meter distance. The access points were set to a constant transmit power. Of the above factors, multipath signal propagation through closed doors was partially limited and one type of mobile device was used. There were 20 tests per each point.

6. are the tests dynamic? What is the accuracy with dynamic tests vs static ones?

Just only static tests were carried out.

7. in the abstract, the authors state that the accuracy of their system is 2.5m but, on the evaluation part there is no information that supports that sentence.

Thank you so much for this comment. It may be a little bit confusing. The 2.5m accuracy mentioned in the abstract is only additional information about our system. The whole concept is inteded as a zone (room) localization system. Zone accuracy was evaluated in the chapter 6.1 (partially edited).

We would like to thank you again and apologize for your time you spent with reviewing of the revised version of our manuscript. Thank you for your contribution.

Reviewer 3 Report

This paper proposed an algorithm based on fuzzy logic to optimize indoor positioning problems. Moreover, the authors also developed a system to show its effectiveness. The resulting average accuracy of positioning is about 2.5 m. However, after carefully reviewing this paper, some technical comments have been raised as follows:

  1. Fonts in Figures are too small to clearly read.

  1. Lines 257-259, all numbers should have the associated units. For example, -60.2 dBm.

  1. Line 304, wrong abbr. was found. FRBS is not the abbr. for fuzzy derivation system.

  1. Line 356, “Optimization of fuzzy sets," Please replace the semicolon with a comma.

  1. The procedure to optimise the FRBS is not clear. What is the input to the fuzzy set? In Figs. 6-8, the x-axis is always x. It is not clear and hard to understand.

  1. The quality of the manuscript is poor. It looks like a _nal project report. In line 426, the IP addr. of the machine is not important at all.

  1. How many training data were used? What is the size of your fingerprinting table? From Table 1, it seems the correct ratio is quite low.

  1. No comparisons with other state-of-the-art at all. The data collection procedure is followed IPIN. Thus, the following papers should be cited.

        Torres-Sospedra et al., “Off-Line Evaluation of Mobile-Centric Indoor    Positioning Systems: The Experiences from the 2017 IPIN Competition", Sensors, 2018, vol. 18, no. 2, pp. 1 - 27.

        The authors can use these open-accessed data to verify your pros of   the proposed algorithm rather than emphesise the implemented system.

Based on the above comments, the reviewer cannot suggest accepting this paper in current form.

Author Response

Dear reviewer,

Thank you so much for your valuable comments and recommendations. We really appreciate time you spent to make a review of our paper. Based on your remarks we have tried to do our best to enrich our article as you suggested. The changes are indicated in an updated version by green color. We really hope that these modifications and addition of some parts will contribute to accept the paper.

1. Fonts in Figures are too small to clearly read.

Thank you so much for your remarks. All figures have been adjusted to increase the font size.

2. Lines 257-259, all numbers should have the associated units. For example, -60.2 dBm.

Thank you so much for your remarks. The corresponding units have been added.

3. Line 304, wrong abbr. was found. FRBS is not the abbr. for fuzzy derivation system.

Thank you so much for your remarks. Terminology corrected (fuzzy rule-based system).

4. Line 356, “Optimization of fuzzy sets," Please replace the semicolon with a comma.

The sentence has been re-phrased.

5. The procedure to optimize the FRBS is not clear. What is the input to the fuzzy set? In Figs. 6-8, the x-axis is always x. It is not clear and hard to understand.

Thank you so much for this comment. It may be a little bit confusing. The input to the fuzzy set (marked as X) is the difference between two RSSIs from two APs and the localization device (e.g. Δ1−2, Δ2−3) as mentioned in the chapter 3.1.

6. The quality of the manuscript is poor. It looks like anual project report. In line 426, the IP addr. of the machine is not important at all.

Thank you so much for your remarks. Manuscript was enriched and IPadress has been removed. We really hope that modifications and addition of some parts will contribute to accept the paper.

7. How many training data were used? What is the size of your fingerprinting table? From Table 1, it seems the correct ratio is quite low.

The determination of the resulting accuracy of the previous localization system was performed using a series of repeated measurements, where the individual points of the database from the training phase were created with a meter distance. The access points were set to a constant transmit power. Of the above factors, multipath signal propagation through closed doors was partially limited and one type of mobile device was used. The measurement itself took place in four stages, each containing forty position calculation intervals.

8. No comparisons with other state-of-the-art at all. The data collection procedure is followed IPIN. Thus, the following papers should be cited.

Torres-Sospedra et al., “Off-Line Evaluation of Mobile-Centric Indoor    Positioning Systems: The Experiences from the 2017 IPIN Competition", Sensors, 2018, vol. 18, no. 2, pp. 1 - 27.

The authors can use these open-accessed data to verify your pros of   the proposed algorithm rather than emphesise the implemented system.

Thank you for this comment. We add new section called related work where we have tried to summarize the state of the art regarding Indoor Positioning Systems and Localization techniques. Based on your recommendations we also did a comparison of these techniques and approaches and listed the main pros and cons.

We would like to thank you again and apologize for your time you spent with reviewing of the revised version of our manuscript. Thank you for your contribution.

Round 2

Reviewer 2 Report

Overall, the authors made a good effort on improving the paper's quality. However, to validate the concept proposed, I would like to recommend the following improvements:

  1. to guarantee the generality of the proposed method, the fuzzy logic must be tested and validated in a different dataset than the one used for the training. 
  2. Dynamic and static conditions impose different challenges on IPS performance. Currently, due to the plethora of IPS available on the literature, an IPS must be evaluated on dynamic conditions. E.g., the setup and configuration of the kalman filter will have to change to adapt to this condition.

Author Response

Dear reviewer,

Thank you so much for your valuable comments and recommendations. We really appreciate time you spent to make a second review of our paper and we are glad that you positively evaluate our first modifications. Based on your remarks we have tried to do our best to enrich our article as you suggested. The changes are indicated in an updated version by blue color. We really hope that these modifications and addition of some parts will contribute to accept the paper.

Overall, the authors made a good effort on improving the paper's quality. However, to validate the concept proposed, I would like to recommend the following improvements:

1. to guarantee the generality of the proposed method, the fuzzy logic must be tested and validated in a different dataset than the one used for the training.

Thank you for this comment. We are aware that it would be appropriate to verify the data on a new data set, however we would like to do this in the following work. As part of the future development of this concept, we plan to refine our solution by verifying the new data set and not only the test data, but we also plan to perform dynamic tests.

2. Dynamic and static conditions impose different challenges on IPS performance. Currently, due to the plethora of IPS available on the literature, an IPS must be evaluated on dynamic conditions. E.g., the setup and configuration of the kalman filter will have to change to adapt to this condition.

Thank you for this recommendation. We add new paragraph to the conclusion where we are mentioning our intention to perform some other tests. The future work should be instead of current fuzzy logic application influence on accuracy devoted to comparison of the results of localization using the Fingerprinting method and evaluate the differences between static and dynamic localization tests. This complex task will require a number of modifications in process methodology, as well as data processing changes such as the different configuration settings of the Kalman filter, etc.

We would like to thank you again and apologize for your time you spent with reviewing of the revised version of our manuscript. Thank you for your help.

Reviewer 3 Report

Thank you for this revision. The quality of the manuscript has been improved significantly. However, some typos were found and something has to be clarified.

  1. Near line 690, "This is true for localization points (3), (4), (7), (9), (10)" should be "...(9) and (10)"
  2. How do the authors define the rate of the correct zone? In environment 2 (Fig. 11), some zones are quite large. In such a case, even the zone is correctly classified. The resulting positioning errors in meters may be large. The authors have to use positioning errors in the distance.

Based on my comments, the reviewer cannot accept your work in the current form.

Author Response

Dear reviewer,

Thank you so much for your valuable comments and recommendations. We really appreciate time you spent to make a second review of our paper and we are glad that you positively evaluate our first modifications. Based on your remarks we have tried to do our best to enrich our article as you suggested. The changes are indicated in an updated version by blue color. We really hope that these modifications and addition of some parts will contribute to accept the paper.

---

Thank you for this revision. The quality of the manuscript has been improved significantly. However, some typos were found and something has to be clarified.

1. Near line 690, "This is true for localization points (3), (4), (7), (9), (10)" should be "...(9) and (10)"

Thank you so much for your remark. The sentence has been re-phrased based on your recommendation.

2. How do the authors define the rate of the correct zone? In environment 2 (Fig. 11), some zones are quite large. In such a case, even the zone is correctly classified. The resulting positioning errors in meters may be large. The authors have to use positioning errors in the distance.

Thank you for this comment. We have tried to more explain, how the zone is created and what a zone is.  As can be seen on Figure 12, the zone is defined as a room. For zone creation, the Zone administration interface is used at FuzzyLoc Supervisor. As mentioned above, the main criterion of the localization system is its accuracy, which depends on the so-called degree of localization level. In this concept, we tried to develop a localization system at the zone level. These are places where there are often a large number of people or medical devices. In the case of managing crisis situations, it is sufficient to have an overview of, for example, how many people are currently in a given zone, etc. A more precise type of localization is not usually required in desired cases.

Based on my comments, the reviewer cannot accept your work in the current form.

---

We would like to thank you again for your time you spent with reviewing of the revised version of our manuscript. Thank you for your help.